# Pilot Testing of Useful Tools’ Validity for Frailty Assessment in Greece: Translated PRISMA-7 Tool, Modified Fried Criteria and Clinical Frailty Scale

**DOI:** 10.3390/healthcare12090930

**Published:** 2024-04-30

**Authors:** George Soulis, Efstathia Kyriakopoulou, Aristea Leventouri, Eleni Zigkiri, Vasiliki Efthymiou, Zikos Kentros, Anastasia Koutsouri

**Affiliations:** 1Outpatient Geriatric Assessment Unit, Henry Dunant Hospital Center, 11526 Athens, Greece; zigkiri@gmail.com (E.Z.); a.koutsouri@dunant.gr (A.K.); 2Hellenic Society for the Study and Research of Ageing, 10677 Athens, Greece; 3Department of Physiotherapy, Henry Dunant Hospital Center, 11526 Athens, Greece; fay_kyr@hotmail.gr (E.K.); z.kentros@dunant.gr (Z.K.); 4Department of Neurology, University General Hospital of Patra, 26504 Patra, Greece; aristinalvn@gmail.com; 5University Research Institute of Maternal and Child Health & Precision Medicine, National and Kapodistrian University of Athens, 11527 Athens, Greece; vikimiou2003@yahoo.gr; 61st Department of Internal Medicine, Henry Dunant Hospital Center, 11526 Athens, Greece

**Keywords:** frailty, Greek translation of PRISMA-7, modified Fried criteria, Greek version of clinical frailty scale

## Abstract

The importance of frailty in older people is getting constant recognition as an important aspect both in terms of public health, as well as at a personal level, for the appropriate management of an older person’s health condition. This is reflected by the continuously increasing number of research studies carried out in several settings across different countries. Sometimes, this is very solid, but in other cases, there is a considerable gap in terms of accurate and well-grounded documentation of frailty status. This is the case in Greece, where we are missing clinically validated tools to approach frailty. We are missing frailty screening tools, such as, for instance, Program of Research on Integration of Services for the Maintenance of Autonomy 7 (PRISMA 7), the gold standard tool of Fried criteria, is somehow problematic since the question referring to physical activity originates from a questionnaire that has not been translated and validated, while Clinical Frailty Scale (CFS) has been validated for translation but not for the capacity to detect frailty. The aim of this study is to validate these tools for their accuracy to detect frailty by using a measurable index of frailty, previously proposed for use in clinical studies: the Short Physical Performance Battery (SPPB). Seventy-four male and female participants (mean age 80.47 years SD = ±7.45 years, minimum–maximum age = 65–95) have been evaluated for their frailty status using different tools. We observed that the PRISMA 7 translation detects frailty only when one question is removed at a cut-off of ≥2 and indicates a sensitivity of 88.1% and specificity of 99.9% with a good correlation with SPPB measurements (*r* = −0.858; *p* < 0.001). When CFS was validated using SPPB, it demonstrated a very good correlation (*r* = −0.838; *p* < 0.001 respectively) as was the case for the modified Fried Criteria (*r* = −0.725; *p* < 0.001). All items demonstrated a good correlation between them. We here propose that we can accurately assess frailty status in the community setting by using a modified version of Fried criteria, Clinical Frailty Scale translation in Greek, and we can screen for frailty by using the Greek translation of PRISMA 7 only after removing item 6 of the questionnaire.

## 1. Introduction

Population aging is rapidly increasing worldwide, while healthcare systems seem inadequate to deal with its multiple consequences. The number of people aged over 65 years old is estimated to reach 2 billion by 2050 [1]. Frailty is a disorder of various inter-related physiological systems, including immune, musculoskeletal, endocrine, cardiovascular, hematopoietic and neural [2]. The process of aging in combination with lifestyle choices, genetic, epigenetic and environmental risk factors, activate chronic inflammation pathways which negatively affect multiple system function, contributing to the development of a frail phenotype [1]. Frailty remains a major challenge, closely associated with advanced age, affecting almost one out of five individuals over 65 years old [3], significantly increasing the risk of falls, disability and long-term care and leading to high morbidity and mortality rates [1]. Its estimated prevalence varies greatly from 4 to 59.1% due to the variety of definitions and criteria used among the different studies [4]. However, frailty rates appear considerably higher in hospitalized geriatric patients and especially in the elderly living in long-term care units [5].

Although the term “frailty” is not new in the field of geriatric medicine, its use is mainly based on clinical experience, and healthcare providers have been unable to accurately define it until recently [2]. According to the World Health Organization, frailty describes an age-related, gradual decline in physiological systems’ function, indicating increased vulnerability to stressors and various adverse health outcomes, as a result of decreased reserves of an individual’s intrinsic capacity. It is a distinct clinical entity, often coexisting but differing from disability or comorbidities [1,6], and it can be prevented, reversed or partially controlled with the appropriate interventions [7]. As a result, frailty’s assessment should definitely be an integral part of the elderly’s primary care, and healthcare providers ought to be adequately qualified on its diagnosis and treatment [8].

Frailty’s pathophysiology is not fully understood. The precise point to which age-related decline in multiple systems and deficit accumulation overcome an organism’s repairing ability and depleted homoeostatic reserves is not yet determined [1]. Two major approaches regarding frailty have been developed: frailty phenotype (FP) and frailty index (FI) [2]. These two models seem to overlap, although FI may have a greater ability to predict frailty’s adverse health outcomes [9]. FP describes frailty as a clinical syndrome, and its diagnosis is based on the presence of at least three of the following five phenotypical criteria: unintended weight loss, muscle weakness, fatigue, slowness and low physical activity. Individuals with the presence of only one or two criteria are considered to be in a prefrail stage, which almost doubles the possibility for the clinical manifestation of frailty in the following 3–4 years [6]. Although FP is concrete and reliable, it does not take into consideration important parameters, such as the presence of cognitive decline. Also, FP serves mainly as a categorical and not a quantitative variable [1]. According to FI, frailty is a state of an age-related progressive deficit accumulation due to a decrease in multiple systems’ physiological reserve [10,11]. FI is a ratio which can be estimated by comparing the values of at least 30 different variables (symptoms and signs, comorbidities, laboratory tests and functional indicators) that may be present in an individual, with the mean values of these variables present in healthy individuals of the same age [10]. A value of 20% is crucial for frailty, whereas a value greater than 67% indicates an increased risk of collapse [1].

According to different studies, obesity or cachexia [2,12], lack of exercise [5], the presence of chronic diseases [6,8], polypharmacy [13], neoplasms [2], stress or depression [9,12], as well as certain demographic characteristics such as advanced age [6,12,14], female gender [2,4,6,14] and African American origin [6] have been associated with an increased risk of frailty. Frailty has also a strong financial and social dimension since low income at a certain life period has been demonstrated to predict the appearance of frailty later in life [15]. On the other hand, raised income, moderate alcohol consumption, elevated health status and high educational level possibly serve as protective factors against frailty or may simply indicate robust individuals [4,12].

It is highly suggested that all patients over 70 years old should be evaluated for the presence of frailty in primary care [3] if we want to promote evidence-based decision making for these people [2,8]. Numerous frailty assessment tools have been developed, although their clinical use is not recommended unless they meet certain criteria: no need of any special equipment, possibility to be completed in less than 10 min, validation and appropriate design for the detection of frailty. The approved diagnostic tools for the initial detection of frailty are the following: Clinical Frailty Scale (CFS), Edmonton Frail Scale (EFS), FRAIL Index, INTER-FRAIL Questionnaire, PRISMA-7 tool, Sherbrooke Postal Questionnaire, Short Physical Performance Battery (SPPB) and Study of Osteoporotic Fractures (SOF) Index [3]. After their clinical application, an individual may be considered robust and so just be encouraged to follow preventive measures and adopt a healthy lifestyle, or the presence of frailty may be detected. In that case, the use of the frailty phenotype or frailty index in addition to a complete geriatric assessment could lead to the definitive diagnosis of frailty requiring multicomponent interventions [3]. The 9-point Clinical Frailty Scale (CFS) remains one of the most widely used tools for the assessment of frailty in clinical practice as it appears simple, rapid and reliable [16]. It is a visual scale that summarizes the overall level of robustness or frailty of older adults, ranging from 1 (very fit) to 9 (terminally ill), taking into consideration their comorbidity, mobility and functional level [11,17]. It is a judgement-based scale, largely depending on clinicians’ experience [18], though with a great prognostic value and with the ability to predict frailty’s possible adverse health outcomes, such as hospitalization or death. Its good diagnostic accuracy can significantly aid frailty’s early identification, thus contributing to a better quality of life [11,17]. Also Fried criteria, based on the frailty phenotype, have been widely used in clinical practice and remain a reference standard in research. However, further studies are required to assess their clinical application that may be limited by the fact that a level of fitness is required to perform the hand grip tests and the walking distance [19]. In addition, the PRISMA-7 (Program of Research to Integrate the Services for the Maintenance of Autonomy-7) yes/no questionnaire was developed in Canada in 2007 and includes seven questions regarding older adults’ age, health status, activities and social support, receiving a score of one or zero. A score ≥ 3 is used as a threshold for frailty. It is a brief and easily applicable tool, and its use is recommended in international guidelines [20].

Despite its clinical and predictive value, frailty has not been adequately investigated in Greece. The aim of this study is to assess the validity of several frailty tools and, more specifically, to translate and validate the PRISMA-7 tool in Greek, to validate the already translated Clinical Frailty Scale in a community-dwelling population and to unlock the use of Fried criteria by producing a modified version of them. Thus, we offer a variety of frailty screening and assessment tools that are easily applicable in everyday clinical practice as well as in research activities.

## 2. Materials and Methods

### 2.1. Design and Setting

This study was cross-sectional and observational. Participants were evaluated at the Outpatient Geriatric Assessment Unit of Henry Dunant Hospital Center in Athens from 22 February 2023 to 31 May 2023. We aimed to recruit male and female participants ≥ 65 years old, especially aiming to include as many participants as possible aged over 85 years old. People included were community dwellers either visiting the outpatient Internal or Geriatric medicine consultation center at Henry Dunant Hospital or volunteers following suggestions from people that had already participated in the study. People with diagnosis of dementia that could not communicate and complete the assessment were excluded. Seventy-four older people participated in the present study. The sample size was determined during the study design according to the bibliography [21,22,23].

### 2.2. Translation

After obtaining official approval from the holders of the copyright of the PRISMA-7 tool, we translated it using the standard steps of establishing an expert committee, proceeding to forward translation by two independent translators, deciding a common text and then moving to backward translation by two independent translators [24]. Instructions accompanying the tool were also translated but only by two separate translators that had concluded to the final text. The translation was then compiled into a final text and compared with the original text. A test re-test reliability survey was carried out in 10 people not included in the core study with 10 days interval between the separate evaluations. Test re-test scores were the same at 10 days interval. The final form (Supplementary File S1: Prisma 7 Translation in Greek) was used for this study. Two experts in this field established face validity.

### 2.3. Statistical Analysis

Demographic data have been recorded for all participants. We evaluated frailty status of participants by using several approaches. The translated PRISMA-7, the Greek translation of Clinical Frailty Scale [25] and the modified Fried criteria were used, while Short Physical Performance Battery was used as an objective tool for measuring frailty status. The study protocol was approved by the institutional Ethics Committee (21 March 2023). All participants signed written informed consent prior to inclusion. This study followed the intentions of the Declaration of Helsinki and the Standard Ethical Principles. Data were analyzed by IBM© SPSS© version 25 (IBM Statistical Package for Social Sciences for Windows, Version 25.0., Armonk, NY, USA: IBM Corp.). The statistical techniques used were descriptive analysis, internal consistency (Kuder–Richardson formula 20 and Cronbach’s alpha), convergent validity (Pearson and Spearman correlation coefficients), factor analysis (Principal Component Analysis with Varimax Rotation Method) and Receiver Operating Characteristic (ROC) analysis. Establishing state variables “Frail” and “Not Frail” from the SPPB test, ROC curves were generated, and the area under the ROC curve (AUC) was used for study outcomes. An index test AUC of at least 0.8 is considered to indicate good discriminative ability [26]. Cohen’s kappa (κ) was conducted in order to check agreement between two categories (frail, non-frail) between Fried criteria and SPPB questionnaires. The level of significance was set at 0.05.

## 3. Results

Seventy-four participants were included in this study. Their sociodemographic characteristics are shown in Table 1. The majority of the participants were female (*n* = 44, 59.5%), while 30 (40.5%) were men. Patients’ mean age was 80.47 years (SD = 7.45 years, minimum–maximum age = 65–95). The age group with highest frequency was the age category of 76–80 years (*n* = 21, 28.4%). The sample was matched for age and sex (*p*-value = 0.298). Moreover, the majority of the participants were married (*n* = 49, 66.2%), graduated from junior/high school (*n* = 31, 41.9%). Almost half of the participants were non-smokers (*n* = 36, 48.6%) and ex-smokers (*n* = 26, 35.1%), and the rest of them were current smokers (*n* = 12, 16.2%). Approximately 1 out of 10 patients had orthostatic hypotension (*n* = 11, 14.9%). The number of chronic conditions ranged between 0 and 6 with mean number 2.80, median 3.00 and standard deviation 1.32.

Construct validity was conducted using factor analysis with principal component extraction method as presented in Table 2. Kaiser–Meyer–Olkin Measure (KMO) of Sampling Adequacy was found as 0.750 and Bartlett’s Test of Sphericity as 158.440 (*p*-value < 0.001). Measures of Sampling Adequacy (MSAs) were acceptable for all items over the value of 0.7 except item 6 (0.282). Low MSA in item 6, [27], led to the removal of this item from the analysis. A new factor analysis was assessed with a higher KMO equal to 0.783. All items now had acceptable MSA. Two components were extracted after Varimax Rotation; both had eigen values over 1 (3.089 and 1.010, respectively, for two components), and the explained variance was 51.48% and 16.83%, respectively (cumulative explained variance for both components was found as 68.31%). Scree plot (Appendix A: Scree plot), Kaiser criterion eigen values which were greater than one [28] and minimum 50% explained variance resulted in the fact that the optimum number of factors was two, as seen in Table 2.

Descriptive statistics of all variables extracted after scoring the tools of the present study are shown in Table 3. Kuder–Richardson formula 20 for the PRISMA scale was found as 0.778.

Using the SPPB test (1–9 score was categorized frail) as a reference (gold) standard, PRISMA 7 showed an excellent discrimination (AUC = 0.915, 95% CI = 0.848–0.981). A cut-off point of 2 or higher for PRISMA was applied and indicated a sensitivity of 88.1% and specificity of 99.9%, as presented in Figure 1.

The convergent validity for the PRISMA Scale and the other administrated tools was adequate as seen in Table 4. More specifically, the greater correlated scale was between PRISMA 7 and SPPB scale as well as CFS (*r* = −0.858; *p* < 0.001 and *r* = −0.838; *p* < 0.001, respectively). In order to check the validity between SPPB and Fried criteria, a high correlation was revealed (*r* = −0.725; *p* < 0.001). Also, Cohen’s kappa (κ) was conducted between frail and non-frail categories in both tools. It was shown that there is statistically significant moderate agreement between them (κ = 0.563, *p* < 0.001), and the respective percentages for frail and non-frail categories were 78.6% and 78.1%.

## 4. Discussion

After translating the PRISMA-7 tool according to ordinary procedures elsewhere described [24], we concluded to the final document of PRISMA-7 questionnaire that was going to be validated. PRISMA-7 is a very useful screening tool at the primary care level that was developed in Canada [29]. We found that the full version of the translated tool does not adequately and accurately describe people that may be classified as frail when assessed by SPPB score lower than 10, which is used as a gold standard tool to characterize baseline physical frailty [30]. Only after removing question 6 which refers to the possibility of having a person that may provide help in case of need (“If you need help, can you count on someone close to you?”), we obtained a good correlation between the estimation of frailty status by PRISMA 7 with a cut-off point of >2 with the measurement of physical frailty (KMO equal to 0.783). This may be due to cultural particularities in Greece that may exist in other countries as well. People from Greece are more likely to report increased support which can be explained by strong family ties and the cultural values of the country [31].

The study of frailty is continuously increasing, but in order to do it accurately, we need validated tools in as many languages possible, adjusted to the real-life circumstances and conditions of each country. Fried criteria are the cornerstone of the available tools used to define frailty but, in some cases, are not applicable, since the original studies used to define some of the aspects that are assessed are not translated and validated in the respective countries. This poses burdens in the study of frailty, since using a reference tool in which some dimensions may not be validated renders a study doubtful. This has already triggered researchers in the field to modify Fried criteria in order to make them usable at the country level [32,33,34]. Considering that the Minnesota Leisure Time Activity questionnaire used to estimate physical inactivity in the original Fried study [6] is not translated or validated in Greek and that there are no other tools specifically designed to evaluate physical activity levels of older adults in order to classify them as active or inactive accordingly, it becomes important to assess physical inactivity by using arbitrary cut-off values and then evaluating their accuracy.

The modification of the Fried criteria that we used concerned Physical Activity (PA), for which we set less than 10 min continuous walk per day as a cut-off for physical inactivity. The International Physical Activity Questionnaire (IPAQ) is designed for people up to 69 years old (IPAQ), and they are using 10 min as the minimum time to count physical activities [35]. This modification seems to accurately and adequately describe the community-based population of the study in terms of recognition of frailty status as estimated with an objective measurement: SPPB score. We understand that PA is a multi-dimensional construct, and thus, there is no measure that can assess all facets of PA [36], but sometimes, we need to be practical. Since our modification works for the community, we propose to use our Modified Fried Criteria as a reference tool to evaluate frailty when performing studies in Greece. It is true that this arbitrary approach may underestimate other physical activities, such as gardening or swimming, but this may be excused considering that the population studied lives in an urban setting where access to a garden is almost impossible, while swimming has a seasonal character in Greece and cannot be considered a yearlong regular physical activity.

The Clinical Frailty Scale translation in Greek has been validated by using the Barthel Index as a comparator, an index that measures the extent to which somebody can function independently and has mobility in their activities of daily living (ADLs) while indicating also the need for assistance in care [37]. We cannot easily claim that Barthel Index describes frailty status. It may describe a part of the frailty status, but it is true that it may be less reliable in patients with cognitive impairment, while it is affected by the degree of disability in the population examined [38]. We decided to evaluate the validity of the CFS translation in Greek by directly comparing it to a tool proposed as a measurement of frailty in clinical studies [30], the Short Physical Performance Battery. Consequently, we compared it with other tools linked with frailty evaluation and screening. It seems that it works well with our community-dwelling population sample, so one can use it to classify the frailty status of a similar population as well.

To our knowledge, this is the first study in Greece trying to validate a translated version of the PRISMA 7 frailty screening tool with frailty status assessed by using the SPPB tool. At the same time, we validated the capacity of the Greek translation of the CFS scale to detect frailty status in the community, and we modified Fried criteria to make them usable according to country specificities and particularities. We compared different frailty tools (screening, measuring, evaluation, classification), and we observed a significant statistical correlation among them.

One major disadvantage of the study is the relatively small size of the sample. It happens in some studies when a tool is used to test a slightly different outcome that an item or few items perform in a poorer than expected way. Our approach to take an existing tool and remove one item poses certain considerations. It is true we lose comparison capacity and develop, in a way, a kind of new tool, and for this, we need a greater sample. So, we are facing the challenge of either modifying the existing PRISMA-7 tool by removing one item and losing certain dimensions of the tool (comparability with other countries, creating a new tool that needs to be evaluated in larger studies)—and this is what we suggest—or exclude the use of this tool in Greece. Another limitation of the study is that the sample was not chosen randomly, but it was somehow selected so that our criteria were met. Existence of practical and validated frailty screening tools is a cornerstone for the detection of frailty even after a short-term training of healthcare professionals, as it has been shown in a feasibility and impact study in Greece, where following a short -erm training resulted in increased awareness in healthcare professionals of the use of screening tools in their everyday practice [39].

As a closing remark, we have to keep in mind that frailty is a complex condition that, most of the times, is difficult to be encapsulated within one definition and one tool, and the clinical experience of the practicing physician is always important. Each of the tools used in this study claims a distinctive position and a valuable role in the evaluation of frailty, especially in the setting and the population that we have chosen. We provide here a great variety of validated tools, each tool for a different context, in order to better approach frailty in Greece depending on the specific needs of the assessment, the setting and the resource availability.

## 5. Conclusions

We demonstrate here an excellent matching between several frailty tools (Greek versions of PRISMA-7, CFS and Modified Fried criteria) and measurements of frailty which were assessed using Short Physical Performance Battery scores as a comparator. We were obliged to exclude one item from the PRISMA-7 questionnaire, due to cultural reasons. The different tools demonstrated excellent correlation among them, a fact that renders them usable and valid for the assessment, characterization and investigation of frailty status in Greece. We propose the use of the Greek version of PRISMA 7 without item 6, which refers to the possibility of having a person that may provide help in case of need, due to cultural particularities. A modified version of Fried criteria proposed here for reasons of practicality seems an accurate and acceptable adaptation that can describe frailty status in the community-dwelling population. Clinical Frailty Scale that has been previously translated and validated for its usability is also an efficient and credible tool for frailty assessment in the same population.

## Figures and Tables

**Figure 1 healthcare-12-00930-f001:**
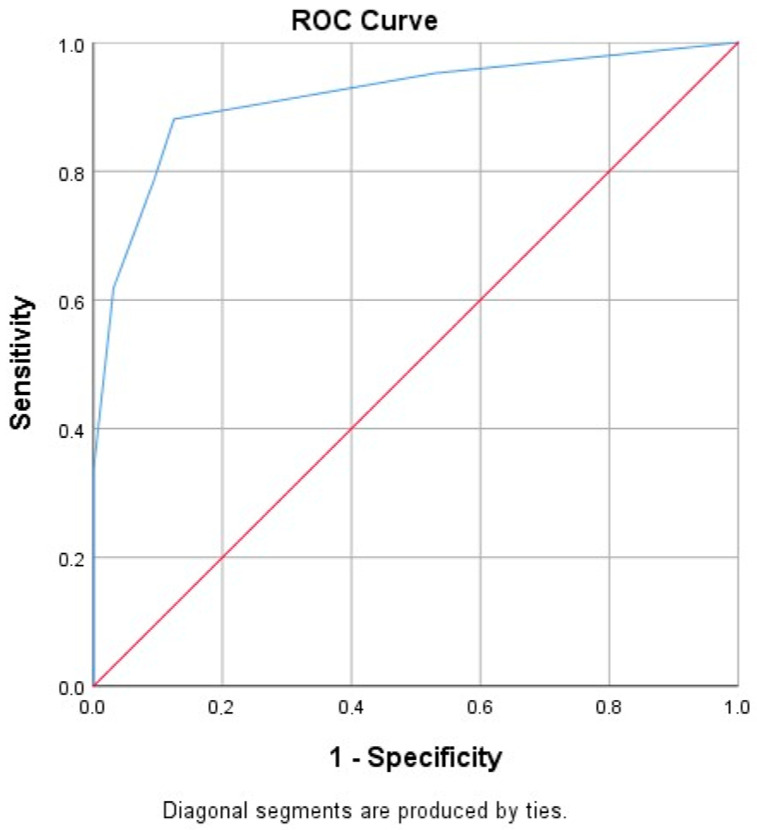
PRISMA 7 Receiver Operating Characteristic (ROC) curve.

**Table 1 healthcare-12-00930-t001:** Sociodemographic characteristics of the participants (N = 74).

Characteristics		*n* (%)
Sex	Male	30 (40.5)
	Female	44 (59.5)
Age (mean ± SD; 80.47 ± 7.45 years)	65–70	8 (10.8)
	71–75	11 (14.9)
	76–80	21 (28.4)
	81–85	13 (17.6)
	86–90	13 (17.6)
	>90	8 (10.8)
Marital status	Single	3 (4.1)
	Married	49 (66.2)
	Widowed	20 (27.0)
	Divorced	2 (2.7)
Educational level	Primary school	16 (21.6)
	Junior/High school	31 (41.9)
	Higher (Technical school)	7 (9.5)
	Highest (University)	16 (21.6)
	Post graduate/PhD	4 (5.4)
Smoking	Never	36 (48.6)
	Ex	26 (35.1)
	Current	12 (16.2)
Orthostatic hypotension	Yes	11 (14.9)
	No	63 (85.1)

**Table 2 healthcare-12-00930-t002:** Factor analysis of PRISMA-7 (N = 74).

Rotated Component Matrix		
	Component	
	1	2
Prisma1	0.675	
Prisma3	0.837	
Prisma4	0.855	
Prisma5	0.727	
Prisma7	0.818	
Prisma2		0.995

Notes. Extraction Method: Principal Component Analysis. Rotation; Method: Varimax with Kaiser Normalization.

**Table 3 healthcare-12-00930-t003:** Frailty measures for the sample (N = 74).

Measures	Mean	SD	Min–Max
PRISMA 7	2.43	2.00	0–6
CFS	3.68	1.07	1–7
FRIED	1.11	1.28	0–5
SPPB	7.89	3.70	1–12

Notes. Values are referred to mean and standard deviation (SD).

**Table 4 healthcare-12-00930-t004:** Convergent validity (N = 74).

Scales	[1]	[2]	[3]	[4]
PRISMA 7 [1]	1			
CFS [2]	0.838 ***	1		
FRIED [3]	0.727 ***	0.765 ***	1	
SPPB [4]	−0.858 ***	−0.826 ***	−0.725 ***	1

Notes. Values are referred to Pearson correlations. *** *p* < 0.001.

## Data Availability

The data presented in this study are available on request from the corresponding author.

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
