# Peer review of "Pilot Testing of Useful Tools’ Validity for Frailty Assessment in Greece: Translated PRISMA-7 Tool, Modified Fried Criteria and Clinical Frailty Scale"

_healthcare, 2024, doi:10.3390/healthcare12090930_

Round 1

Reviewer 1 Report

Comments and Suggestions for Authors

Title:

“Pilot Testing of Useful Tools Validity for Frailty Assessment in Greece: Translated PRISMA-7 Tool, Modified Fried Criteria and Clinical Frailty Scale.”

Review

The manuscript proposes a new measure of frailty based on different tools. The objective of this review is to try to improve this study. Some suggestions could be indicated.

1. The participation rate of the study could be mentioned, considering the people who participated and the total invited to participate.

2. The characteristic of the participants may be indicated with a more detail in relation to health status and diseases in order to know better the study population.

3. There are some inconsistencies in the correlations between abstract, Tables and text. Examples

-PRISMA-7 and SPPB in the abstract correlation 0.858, in Table 4 correlation -0.858, and 0.858 in text (Page 6, line 222)

-PRISMA -7 and CFS in the abstract correlation -0.838, and in Table 4, correlation 0.838, and -0.838 in text (Page 6, line 222).

4. The authors could mention the value of Cohen´s kappa as moderate (0.563) (Page 7, paragraph 1), following the guidelines of Landis JR and Koch GG: Biometric 1977;33:159-174.  

 5. A limitation of this study may be that the sample was not being chosen at random in the population of geriatric outpatients in the Henry Dunant Hospital. 

6. In the abstract, CFS can be indicated after Clinical Frailty Scale.

7. The references should follow the journal guideline.

Author Response

Dear Reviewer,

We would like to thank the reviewers for their contribution to the amelioration of the manuscript. We have also performed some minor editing in order to improve the linguistic part of the text.

We are providing our feedback to their comments.

Reviewer 1

The manuscript proposes a new measure of frailty based on different tools. The objective of this review is to try to improve this study. Some suggestions could be indicated.

Thank you for your valuable input.

1. The participation rate of the study could be mentioned, considering the people who participated and the total invited to participate.

This is a very relevant suggestion, but given the fact that the recruitment was not designed to be performed in a consecutive way, but we aimed to obtain a certain number of participants at specified age groups. When we completed a certain number of people aged >65 to 75 we aimed to recruit people of older age and we omitted people of the previous age group and this was not recorded or calculated.

2. The characteristic of the participants may be indicated with a more detail in relation to health status and diseases in order to know better the study population.

In the results section it is added “The number of chronic conditions ranged between 0 to 6 with mean number 2.80, media 3.00 and standard deviation 1.32”

3. There are some inconsistencies in the correlations between abstract, Tables and text. Examples

-PRISMA-7 and SPPB in the abstract correlation 0.858, in Table 4 correlation -0.858, and 0.858 in text (Page 6, line 222)

-PRISMA -7 and CFS in the abstract correlation -0.838, and in Table 4, correlation 0.838, and -0.838 in text (Page 6, line 222).

Thank you very much for this important and difficult to find remark. It is now changed to the text to the correct (r= -0.858; p < 0.001 and r= 0.838; p< 0.001 respectively).

4. The authors could mention the value of Cohen´s kappa as moderate (0.563) (Page 7, paragraph 1), following the guidelines of Landis JR and Koch GG: Biometric 1977;33:159-174.

In the text we added moderate statistically significant agreement between them... instead of statistically significant agreement between them. (line 231)

5. A limitation of this study may be that the sample was not being chosen at random in the population of geriatric outpatients in the Henry Dunant Hospital.

You are right. We added this comment to the discussion part

6. In the abstract, CFS can be indicated after Clinical Frailty Scale.

We added (CFS) after Clinical Frailty Scale in the abstract

7. The references should follow the journal guideline.

Unfortunately we did not understand this comment and we suppose that this is going to be done during the editing process.

Kind Regards

Reviewer 2 Report

Comments and Suggestions for Authors

This study investigated a Pilot Testing of Useful Tools Validity for Frailty Assessment in Greece.

The study, which proposes that, in the community setting, frailty status can be accurately assessed by using a modified version of Fried criteria, Clinical Frailty Scale translation in Greek and can screen for frailty by using the Greek translation of PRISMA 7, is interesting. 

Nevertheless, I have some comments and suggestions for authors, listed below: 

Major comments

Lines 40-42: “Population aging is rapidly increasing worldwide, while healthcare systems seem inadequate to deal with its multiple consequences. The number of people aged over 65 years old is estimated to reach 2 billion by 2050 [1]. Frailty remains…”

Before explanations about frailty, the authors should give some information about the different weakening in aging (cognitive, perceptive…).

Minor comments (comments on the whole manuscript)

Line 23: “…while Clinical Frailty Scale…” : the authors has to add the acronym.

ð “… while Clinical Frailty Scale (CFS)…” 

Line 26: “. 74 male…” : A number cannot be written in the beginning of a sentence.

ð “Seventy-four male…” 

Line 122: “…includes 7 questions…”: numbers until 9 have to be written in letters.

ð “…includes seven questions…”

Lines 143-146: “The sample size was determined during the study design following three criteria; i) 51 more cases than the number of variables [21] ii) at least 10 cases for each item and subjects-to variables [STV] ratio no lower than 5 [22],iii) at least 100 cases and a STV ratio of no less than 5 [23].

The authors should use different punctuation.

ð The sample size was determined during the study design following three criteria: i) 51 more cases than the number of variables [21]; ii) at least 10 cases for each item and subjects-to variables [STV] ratio no lower than 5 [22]; iii) at least 100 cases and a STV ratio of no less than 5 [23].

Author Response

Dear Reviewer,

We would like to thank the reviewers for their contribution to the amelioration of the manuscript. We have also performed some minor editing in order to improve the linguistic part of the text.

We are providing our feedback to their comments.

Reviewer 2

The study,which proposes that, in the community setting, frailty status can be accurately assessed by using a modified version of Fried criteria, Clinical Frailty Scale translation in Greek and can screen for frailty by using the Greek translation of PRISMA 7, is interesting.

Nevertheless, I have some comments and suggestions for authors, listed below:

Thank you for your comments

Major comments

Lines 40-42: “Population aging is rapidly increasing worldwide, while healthcare systems seem inadequate to deal with its multiple consequences. The number of people aged over 65 years old is estimated to reach 2 billion by 2050 [1]. Frailty remains…”

Before explanations about frailty, the authors should give some information about the different weakening in aging (cognitive, perceptive…).

We have changed as suggested our introducton by moving a part of the text to the proposed place, in order to make more sense. Citations have been changed accordingly

Now it reads:

Frailty is a disorder of various inter-related physiological systems, including immune, musculoskeletal, endocrine, cardiovascular, hematopoietic and neural [2]. The process of aging in combination with lifestyle choices, genetic, epigenetic and environmental risk factors, activate chronic inflammation pathways which negatively affect multiple system function, contributing to the development of a frail phenotype [1] Frailty remains a major challenge, closely associated with advanced age, affecting almost one out of five individuals over 65 years old [3], significantly increasing the risk of falls, disability and long-term care, and leading to high morbidity and mortality rates [1].

Minor comments (comments on the whole manuscript)

Line 23: “…while Clinical Frailty Scale…” : the authors has to add the acronym.

ð“… while Clinical Frailty Scale (CFS)…”

We added the acronym as suggested.

Line 26: “74 male…” : A number cannot be written in the beginning of a sentence.

ðSeventy-four male…”

We apologize for that. It is changed according to your suggestion.

Line 122: “…includes 7 questions…”: numbers until 9 have to be written in letters.

ð“…includes seven questions…”

We apologize for that as well. It is corrected

Lines 143-146: “The sample size was determined during the study design following three criteria; i) 51 more cases than the number of variables [21] ii) at least 10 cases for each item and subjects-to variables [STV] ratio no lower than 5 [22],iii) at least 100 cases and a STV ratio of no less than 5 [23].

The authors should use different punctuation.

ðThe sample size was determined during the study design following three criteria: i) 51 more cases than the number of variables [21]; ii) at least 10 cases for each item and subjects-to variables [STV] ratio no lower than 5 [22]; iii) at least 100 cases and a STV ratio of no less than 5 [23].

Another mistake that thanks to your input. It is now corrected to The sample size was determined during the study design according to bibliography [21], [22] [23].

Kind Regards

Reviewer 3 Report

Comments and Suggestions for Authors

This was a very interesting, high-quality read that would certainly be of interest to Healthcare readers. I only have a couple of suggestions.

These are my comments to the respected authors.

Line 202 – The authors mention that a scree plot is provided in the supplementary file. Possibly due to oversight, the scree plot is not among the downloadable material. Please, add it.

Lines 269-271 – The authors state “It is true that this arbitrary approach may underestimate other physical activities, such as gardening, but this may be quenched considering that the population studied lives in an urban setting where access to a garden is almost impossible.” This seems like a sound explanation, but only if one thinks about gardening as an example. There are certainly other activities more easily available to urban area dwelling old that could not be as readily dismissed as gardening. In that case, this aspect should be further explained.

Sincerely,

The reviewer

Author Response

Dear Reviewer,

We would like to thank the reviewers for their contribution to the amelioration of the manuscript. We have also performed some minor editing in order to improve the linguistic part of the text.

We are providing our feedback to their comments.

Reviewer 3

Comments and Suggestions for Authors

This was a very interesting, high-quality read that would certainly be of interest to Healthcare readers. I only have a couple of suggestions.

We appreciate your supportive comments.

These are my comments to the respected authors.

Line 202 – The authors mention that a scree plot is provided in the supplementary file. Possibly due to oversight, the scree plot is not among the downloadable material. Please, add it.

It is now added.

Lines 269-271 – The authors state “It is true that this arbitrary approach may underestimate other physical activities, such as gardening, but this may be quenched considering that the population studied lives in an urban setting where access to a garden is almost impossible.” This seems like a sound explanation, but only if one thinks about gardening as an example. There are certainly other activities more easily available to urban area dwelling old that could not be as readily dismissed as gardening. In that case, this aspect should be further explained.

We have also commented also on the swimming as a physical activity.

Kind Regards

Reviewer 4 Report

Comments and Suggestions for Authors

After my suggestions have been incorporated, the paper can be published. 

Title: Pilot Testing of Useful Tools Validity for Frailty Assessment in Greece: Translated PRISMA-7 Tool, Modified Fried Criteria and Clinical Frailty Scale.

The review offers feedback designed to augment the submissions quality.

Line 21: Prior to using and abbreviation, author should state the full meaning of the abbreviation. That applies to the whole manuscript.

Line 26: 76 male and female.. please specify their age.

Lines from 76 -80 should go to Line 58 or above where you discuss term frailty.

Line 93-95: These sentences lacks clarity and demands rephrasing.

Line 145-147:  i)..... requires rewording

Line 230: In the Discussion it would be advisable to compare, differentiate between PRISMA-7, CFS, Modified Fried criteria and Short Physical Performance Battery scores, and its strong sides and limitations.

Each of these tools plays a valuable role in identifying frailty and functional decline in older cohorts, with the choice of tool depending on the specific needs of the assessment, setting, and the resources available, and that should be explicitly define.

Comments on the Quality of English Language

Textual revisions are needed. 

Author Response

Dear Reviewer,

We would like to thank the reviewers for their contribution to the amelioration of the manuscript. We have also performed some minor editing in order to improve the linguistic part of the text.

We are providing our feedback to their comments.

Reviewer 4

Title: Pilot Testing of Useful Tools Validity for Frailty Assessment in Greece: Translated PRISMA-7 Tool, Modified Fried Criteria and Clinical Frailty Scale.

The review offers feedback designed to augment the submissions quality.

We thank you as well for your contribution to the improvement of our work presentation.

Line 21: Prior to using and abbreviation, author should state the full meaning of the abbreviation. That applies to the whole manuscript.

It is done.

Line 26: 76 male and female.. please specify their age.

It is done as suggested.

Lines from 76 -80 should go to Line 58 or above where you discuss term frailty.

Thank you for this suggestion that makes the introductory text more cohesive. It is done.

Line 93-95: These sentences lacks clarity and demands rephrasing.

You are right.

Not it is rephrased from

Frailty’s major clinical significance includes risk assessment and prognosis 93 estimation for community-dwelling or hospitalized older adults, leading to evidence-94 based decision making [5, 8]. It is highly suggested that all patients over 70 years old 95 should be evaluated for the presence of frailty in primary care [

to

It is highly suggested that all patients over 70 years old should be evaluated for the presence of frailty in primary care [2] if we want to promote evidence-based decision making for these people [5, 8].

Line 145-147: i)..... requires rewording

Now it appears

The sample size was determined during the study design according to bibliography [21], [22] [23].

instead of The sample size was determined during the study design following three criteria; i) 51 more cases than the number of variables [21]; ii) at least 10 cases for each item and subjects-to-variables [STV] ratio no lower than 5 [22]; iii) at least 100 cases and a STV ratio of no less than 5 [23].

Line 230: In the Discussion it would be advisable to compare, differentiate between PRISMA-7, CFS, Modified Fried criteria and Short Physical Performance Battery scores, and its strong sides and limitations.

Each of these tools plays a valuable role in identifying frailty and functional decline in older cohorts, with the choice of tool depending on the specific needs of the assessment, setting, and the resources available, and that should be explicitly define.

We now have added a closing remark based on your suggestion and we thank you for that

As a closing remark we have to keep in mind that frailty is a complex condition that most of the times is difficult to be encapsulated within one definition, one tool and the clinical experience of the practicing physician is always important. Each of the tools used in this study claims a distinctive position and a valuable role in the evaluation of frailty especially in the setting and the population that we have chosen. We provide here a great variety of validated tools, each tool for different context, in order to better approach frailty in Greece depending on the specific needs of the assessment, the setting and the resource availability.

Kind Regards
